# Prevention of Rat Intestinal Injury with a Drug Combination of Melatonin and Misoprostol

**DOI:** 10.3390/ijms21186771

**Published:** 2020-09-15

**Authors:** David Dahlgren, Maria-José Cano-Cebrián, Per M. Hellström, Alkwin Wanders, Markus Sjöblom, Hans Lennernäs

**Affiliations:** 1Department of Pharmaceutical Biosciences, Uppsala University, 752 37 Uppsala, Sweden; david.dahlgren@farmbio.uu.se; 2Department of Pharmacy, Pharmaceutical Technology and Parasitology, University of Valencia, 46010 Valencia, Spain; Maria.Jose.Cano@uv.es; 3Department of Medical Sciences, Gastroenterology/Hepatology, Uppsala University, 751 85 Uppsala, Sweden; per.hellstrom@medsci.uu.se; 4Department of Pathology, Aalborg University Hospital, 9100 Aalborg, Denmark; alkwin.wanders@rn.dk; 5Department of Neuroscience, Uppsala University, 751 24 Uppsala, Sweden; Markus.Sjoblom@neuro.uu.se

**Keywords:** intestinal barrier dysfunction, single-pass intestinal perfusion, intestinal permeability, gastrointestinal physiology

## Abstract

A healthy intestinal barrier prevents uptake of allergens and toxins, whereas intestinal permeability increases following chemotherapy and in many gastrointestinal and systemic diseases and disorders. Currently, there are no approved drugs that target and repair the intestinal epithelial barrier while there is a medical need for such treatment in gastrointestinal and related conditions. The objective of this single-pass intestinal perfusion study in rats was to investigate the preventive cytoprotective effect of three mucosal protective drugs—melatonin, misoprostol, and teduglutide—with different mechanisms of action on an acute jejunal injury induced by exposing the intestine for 15 min to the anionic surfactant, sodium dodecyl sulfate (SDS). The effect was evaluated by monitoring intestinal clearance of ^51^Cr-labeled ethylenediaminetetraacetate and intestinal histology before, during, and after luminal exposure to SDS. Our results showed that separate pharmacological pretreatments with luminal misoprostol and melatonin reduced acute SDS-induced intestinal injury by 47% and 58%, respectively, while their use in combination abolished this injury. This data supports further development of drug combinations for oral treatments of conditions and disorders related to a dysregulated or compromised mucosal epithelial barrier.

## 1. Introduction

The role of the intestinal mucosa is to form a selective and dynamic barrier between the external luminal contents and the underlying tissue and systemic circulation [1]. It should restrict passage of potentially harmful intestinal constituents, such as microbiota, toxins, and allergens, while allowing carrier-mediated and/or passive transport of water, nutrients, and ions. The mucosal barrier consists of a single layer of intestinal epithelial cells (IEC) covered with a mucus layer and the underlying immune system. The IECs are sealed together at the apical surface by tight junction proteins, which form the primary physical barrier to transport of small hydrophilic molecules across the epithelium. Permeation across this barrier is strictly regulated by a range of neuroendocrine processes, hormones, and luminal stimuli that jointly uphold homeostasis [2,3]. The IECs have a high turnover, being renewed by cryptal stem cells every five days and shredded at the villus tip. This process takes place without any loss of intestinal barrier function in a healthy, selective mucosal barrier [4].

A dysregulated or compromised intestinal barrier permits uptake of potentially toxic xenobiotics and alters absorptive and secretory physiological functions. This may cause local tissue injury, including inflammation, as well as extraintestinal manifestations after permeation and spread of noxious compounds throughout the body. Increased intestinal permeability occurs as a result of chemotherapy and radiation as well as in a range of gastrointestinal (GI) and systemic diseases and disorders, such as celiac disease, inflammatory bowel disease, type 1 diabetes, obesity, nonalcoholic fatty liver disease, alcoholic liver disease, and irritable bowel syndrome [5,6,7,8]. However, a true association between increased intestinal permeability and different disease entities is difficult to define as it requires a cause and effect analysis through studies of temporal relationships. It is yet unproven whether reinforcement of the intestinal barrier can prevent or cure GI or systemic clinical manifestations as there are no approved drugs that target the intestinal epithelial barrier. This opens the possibility for a new pharmaceutical therapeutic approach.

One potential target for such treatment is the melatonin membrane G protein-coupled MT1 and MT2 receptors, which are expressed throughout the GI tract of rats and humans [9,10,11]. Melatonin is a serotonin derivative synthesized by the enterochromaffin cells in the intestine. It reduces basal paracellular permeability in the duodenum through an inhibitory nicotinic receptor-mediated neural pathway in rats [12]. Melatonin has also been shown to reduce ethanol- and radiation-induced increases of intestinal permeability and injury in rats [13,14].

Mucosal homeostasis and inflammation are also mediated by prostaglandins, which have a host of complex long- and short-term effects on the intestine, including regulation of bicarbonate secretion, mucus production, and mucosal blood flow. Misoprostol is a synthetic E-type prostaglandin analogue with cytoprotective actions in the intestine, partly mediated by inhibited production and release of cytokines [15]. Misoprostol is used for preventing GI mucosal perturbation and ulcers induced by nonsteroidal anti-inflammatory drugs [16].

Teduglutide, a glucagon-like peptide-2 (GLP-2) analogue, promotes intestinal mucosal growth. It is indicated for the treatment of short bowel syndrome because it increases intestinal nutrient and fluid absorption and reduces the need for parenteral support. Teduglutide also protects intestinal stem cells from radiation damage in mice as well as acid-induced esophageal mucosal damage in rats [17,18].

The main objective of our single-pass intestinal perfusion (SPIP) study in rats was to investigate the mucosal preventive cytoprotective effect of melatonin, misoprostol, and teduglutide, each with a different mechanism of action. Intestinal mucosal injury was induced by exposing the intestinal mucosa to the anionic surfactant, sodium dodecyl sulfate (SDS), for 15 min. This surfactant causes an acute general mucosal injury manifested by an increase in paracellular and transcellular permeability, coupled with histological changes and leakage of intracellular components [19,20]. The study drugs were administered intravenously (IV) and/or luminally before and/or during the SDS exposure. Mucosal preventive cytoprotective effects of the drugs were evaluated by monitoring intestinal passive transport of ^51^Cr-labeled ethylenediaminetetraacetate (^51^Cr-EDTA)—an inert market for mucosal barrier integrity—and evaluation of histological changes before, during, and after luminal exposure to SDS [21].

## 2. Results

### 2.1. Blood-to-Lumen ^51^Cr-EDTA clearance (CL_Cr-EDTA_) over Time Curves, CL_AUC,_ and CL_max_

The mean (±SEM) CL_Cr-EDTA_ over time (0–120 min) for the nine different experimental designs (see Table 2 and Figure 5 in the method section) are presented in Figures 1–3. The corresponding CL_AUC_ and CL_max_ values calculated from these CL_Cr-EDTA_ time curves (45–120 min) are presented in Table 1.

### 2.2. Effect of Luminal SDS with or without Intravenous Parecoxib or Teduglutide

The CL_AUC_ and CL_max_ from the CL_Cr-EDTA_ time curves (45–120 min) were significantly higher than for the control solution when SDS was added to the intestinal lumen between 45 and 60 min. Parecoxib alone had no effect on the SDS-induced increase in CL_Cr-EDTA_ (Table 1 and Figure 1a). A teduglutide bolus followed by infusion (SDS TED-IV) did not affect the SDS-induced increase in CL_AUC_ and CL_max_ (Table 1 and Figure 1b).

### 2.3. Effect of Intravenously and Luminally Perfused Melatonin

A melatonin bolus at 30 min (SDS MEL-IV) did not affect the luminal SDS-induced increase in CL_AUC_ and CL_max_ from the CL_Cr-EDTA_ time curves (45–120 min) but resulted in a 15 min delay in the SDS-induced increase in CL_Cr-EDTA_ (Table 1 and Figure 2a). Luminal addition of melatonin to the control (Control MEL-LUM) made no difference in CL_AUC_ and CL_max_ but significantly decreased the SDS-induced CL_AUC_ (51%) and CL_max_ (50%), (Table 1 and Figure 2b).

### 2.4. Effect of Luminally Perfused Misoprostol

Luminal misoprostol (SDS MIS-LUM) significantly reduced the SDS-induced increase in CL_AUC_ (38%) and CL_max_ (33%) from the CL_Cr-EDTA_ time curves (45–120 min), (Table 1 and Figure 3a). Luminal misoprostol also resulted in a complete return to baseline CL_Cr-EDTA_ at the end of the recovery period.

### 2.5. Combination Effect of Luminal Melatonin and Misoprostol 

Luminal misoprostol and melatonin in combination (SDS MIS-MEL-LUM) completely inhibited the SDS-induced increase in CL_AUC_ and CL_max_ from the CL_Cr-EDTA_ time curves (45–120 min) (Table 1 and Figure 3b). The combination of melatonin and misoprostol also significantly decreased CL_AUC_ and CL_max_ compared to administration of them separately (Table 1).

### 2.6. Histology

The histological investigations displayed no short-term disruption or detachment of epithelial cells. Neither were there any signs of acute damage or acute inflammation (as determined by the absence of neutrophils). The only feature that differed between the study groups was a clear edema in the tip of several villi in the two experimental groups (misoprostol and misoprostol + melatonin) exposed to misoprostol for 60 min. This was easily seen even with low microscopic magnification (high magnification is shown in Figure 4a,b). The mucus layer was intact in all animals without any obvious damage (see Figure 4c). 

## 3. Discussion

This study investigated the potential of treating an injured intestinal barrier, which is common in many GI and systemic diseases [5]. The main objective was to investigate the protective effects and recovery time following an acute jejunal injury in rats induced by intrajejunal SDS exposure for 15 min in a single-pass intestinal perfusion model [12,22]. Three drugs—teduglutide, misoprostol, and melatonin—with different mechanisms of action were administered by different routes alone and in combination. Their effects were investigated by monitoring CL_Cr-EDTA_ excretion into the jejunum over time and by evaluating histology on the same intestinal segment.

Injury or dysregulation of the epithelial barrier may result in an increased intestinal permeability [5,6,7]. The medical potential is therefore considerable for drug treatment strategies directly targeting recovery of the epithelial barriers. Such drug treatments may halt disease progression in patients with conditions related to a compromised intestinal barrier as well reduce GI-related side effects following chemotherapy. The mechanism of SDS-induced injury is related to its surfactant properties—monomers are incorporated into the epithelial bilayer, causing destabilization of the membrane and the tight junction protein complex [23,24,25]. This increases the passive intestinal transport of a range of compounds in both absorptive and secretory directions and blocks constitutive endocytosis [20,26,27]. The effects on intestinal permeability after exposure to SDS are both concentration- and time-dependent and accompanied by biochemical and histological features of intestinal injury [19,22]. Other common rodent models for studying epithelial injury are the ischemia–reperfusion model; the use of knockout mice lacking vital paracellular junction proteins; and chemical agents inducing epithelial damage, such as bile salt, trinitrobenzene sulfonic acid, acetic acid, or dextran sodium sulfate [28,29,30,31,32]. The similarity to SDS in acute effects on the mucosal barrier in these models makes us confident that SDS can be used to investigate the dynamic barrier-protective effect of a range of drugs. 

Pretreating rats with a selective COX-2 inhibitor, such as parecoxib, has been shown to restore normal intestinal physiology following laparotomy as the surgery itself causes postoperative intestinal paralysis, partly mediated by COX-2-derived intestinal prostacyclin [33]. The paralysis attenuates intestinal functions, such as motility, alkaline secretion, osmoregulation, and water transport, which may all affect the relevance of the data [34]. Furthermore, parecoxib has been previously shown to have no effect on basal intestinal permeability in rats [35] and had no effect on SDS-induced epithelial injury in our present study. Therefore, parecoxib was included in all other study groups to permit optimal physiological conditions for the SPIP model. 

Teduglutide was developed for the treatment of short bowel syndrome. It stimulates intestinal growth and inhibits cell apoptosis manifested as an increase in tissue weight, villus height, and crypt depth [36]. In humans, teduglutide (0.03–0.20 mg/kg/day subcutaneously for >8 weeks) reduces the need for parenteral nutrient support in short bowel syndrome and has shown promising results for treatment of Crohn’s disease [37,38]. Similar dosing for one week results in protective effects on the mucosa and less radiation-induced damage of the small intestine in rats and acid-induced tracheal damage in mice [17,18] Furthermore, jejunal epithelial permeability of ^51^Cr-EDTA is reduced as early as 4 h upon postdosing 5 μg of another GLP-2 analogue (h[Gly 2]GLP-2) in mice and an increased intestinal blood flow from 1 h after administration (0.9 nmol/kg) in rats [39,40]. However, our study could not verify any short-term protective effects (from 45 min before SDS exposure) on mucosal injury after teduglutide (100 μg bolus plus 33 μg/h IV infusion) during 120 min. Hence, additional drug combination experiments, including teduglutide, were not investigated in this acute intestinal injury model. However, further studies on the mucosal protective effects of GLP-2 analogues are warranted in which longer treatment periods should be investigated.

Misoprostol is an agonist for the G-protein-coupled prostaglandin E receptors 1–4 [41]. These receptors are involved in the epithelial homeostasis and protect against mucosal damage; misoprostol is therefore used for the prevention of nonsteroidal anti-inflammatory drug-induced mucosal erosions and ulcers [16,42]. It does so by regulating gastric acid secretion, mucus secretion, and proinflammatory cytokine production and by activating adaptive cell survival pathways through selective gene repression and splicing [15,43]. Misoprostol, being a prostaglandin analogue, also induces some early signs of inflammation, such as increased mucosal blood flow and edema, as confirmed with histology in our study [44]. Accordingly, in the rat intestinal instillation model, pretreating the colon with misoprostol (1 μM) resulted in a 24% reduction in sodium caprate-induced plasma exposure of a permeability probe, FITC-dextran [45]. In comparison, our study of the rat jejunum that used misoprostol (10 μM) resulted in a 50% reduction of the SDS-induced increase in luminal exposure of ^51^Cr-EDTA. The abundant preclinical and clinical data supporting the cytoprotective effects of misoprostol makes it a promising drug for further investigation and treatment of an injured epithelial barrier.

Melatonin has shown positive treatment effects in irritable bowel syndrome and inflammatory bowel disease [46,47]. It stimulates duodenal bicarbonate secretion and epithelial barrier function in rats and mitigates ischemia-, chemical-, and radiation-induced intestinal damage in mice [12,14,31,48]. In rats, both long- and short-term oral and intestinal melatonin attenuate ethanol-induced increases in duodenal permeability by 50% via a nicotinic receptor-mediated pathway [13,49]. This corresponds well to the 50% reduction in SDS-induced damage observed in our study, suggesting that the type of mucosal injury by ethanol and SDS is similar. This is in contrast to the acute duodenal injury induced by 50 mM hydrochloric acid, which is unaffected by melatonin [13]. This shows that different types of epithelial barrier injury are mediated via different mechanisms. The effect of melatonin may also differ between intestinal segments, as illustrated by the lack of effect by melatonin in the jejunum compared to the duodenum [12]. Similar regional intestinal differences may also explain why IV melatonin was found to be effective at reducing duodenal, but not jejunal, injury in another study [13]. In summary, melatonin has a potent effect on mitigating mucosal injury in the jejunum. This result calls for further in vivo investigation of the mechanism and how it contributes to mucosal health and barrier regulation in other intestinal injury and disease models.

The cytoprotective effect following separate administration of misoprostol and melatonin reported by others, and verified by us, encouraged us to evaluate their combined local effect [50]. This resulted in a complete abolishment of the SDS-induced increase in CL_Cr-EDTA_. A combined treatment acting on the mucosal epithelium through different mechanisms may help evade acute mucosal damage. It remains to be investigated if this also holds true for other drug combinations and in other disease-relevant experimental models. 

Previous in vivo histological investigations following small intestinal exposure of SDS at 10 mg/mL in rats showed villus shortening, erosion, and eruption [19,51]. These reported effects were evident directly following a 60 min perfusion and 15 min after an oral bolus, with partial or complete recovery 30 min post the oral bolus dosing. Our study showed increased CL_Cr-EDTA_ when SDS (5 mg/mL) was perfused for 15 min, which suggests epithelial injury [52]. However, the histological examinations of the jejunal segments revealed no acute mucosal changes. Similar trends have been reported by others; the substantial effects on CL_Cr-EDTA_ induced by 30 min luminal exposure to 15% ethanol does not show any histological changes [13]. Furthermore, there is no recovery in CL_Cr-EDTA_ following 60 min luminal exposure of 5 mg/mL SDS, indicating that higher concentration and longer exposure time result in more profound, and most likely permanent, mucosal changes [22]. Consequently, the relationship between the luminal SDS concentration and exposure time to CL_Cr-EDTA_ and histological changes is not obvious. It needs to be investigated if melatonin and misoprostol reduce potential epithelial injury observed at longer SDS exposure times.

Our present in vivo model of epithelial barrier function will be used to improve understanding of chemotherapy-induced mucositis [8,53,54,55]. The rapid proliferation of IECs, together with the complex immunological role and interactions with the gut microbiota, makes the GI tract particularly vulnerable to the tissue-damaging effects of chemotherapeutics. The inability to rapidly repair and restore epithelial barrier function is detrimental to cancer patients as it can result in various pathologies, including sepsis and multiple organ dysfunction and failure. 

In conclusion, our single-pass jejunal perfusion study showed that separate pharmacological treatments with luminal misoprostol and melatonin reduced acute SDS-induced intestinal injury, while their use in combination exerted a profound preventive effect by abolishing the injury. These data support further development of drug combinations for oral treatment of conditions and disorders related to a dysregulated or compromised mucosal epithelial barrier.

## 4. Materials and Methods

### 4.1. Materials

Melatonin, SDS, ethanol, accustain (formalin solution 10% neutral buffered), and Inactin (thiobutabarbital) were purchased from Sigma-Aldrich (St. Louis, MO, USA). Misoprostol was purchased from Tocris Bioscience (Bristol, UK). Teduglutide was purchased from BOC Sciences (Shirley, NY, USA). Sodium phosphate dibasic dihydrate (Na_2_HPO_4_·2H_2_O), potassium dihydrogen phosphate (KH_2_PO_4_), sodium hydroxide (NaOH), and sodium chloride (NaCl) were purchased from Merck KGaA (Darmstadt, Germany). ^51^Cr-EDTA was purchased from PerkinElmer Life Sciences (Boston, MA, USA). Parecoxib (Dynastat) was obtained from Apoteket AB, Uppsala, Sweden. 

### 4.2. Study Formulations

Two isotonic (290 mOsm) phosphate-buffered (pH 6.5, 8 mM) perfusate solutions were prepared: one as a control and the other with SDS (5 mg/mL). These perfusate solutions were investigated with and without added melatonin (100 μM) and/or misoprostol (10 μM) and following IV administration of saline solutions of teduglutide (100 μg + 33 μg/h), melatonin (20 mg/kg), and/or parecoxib (10 mg/kg). Ethanol stock solutions of melatonin (15 mg/mL) and misoprostol (3.3 mg/mL) were added to the perfusate solutions with final ethanol concentrations always below 0.5%. Osmolarity was determined by freezing-point decrement using a Micro Osmometer (Model 3MO; Advanced Instruments, Needham Heights, MA, USA). 

### 4.3. Animals, Anesthesia, and Surgery

The surgical procedure and experimental setup of the rat SPIP experiment has been previously described [20]. The study was approved by the local ethics committee for animal research (no. C64/16) in Uppsala, Sweden. In short, male Han Wistar rats (strain 273) from Charles River Co. (Germany), body weight 270–395 g, were used. The animals arrived at the animal lab facility at least one week before the experiment and were allowed water and food ad libitum during this period. Housing conditions were 12:12 h light–dark cycle and 21–22 °C. On the study day, the rats were anesthetized using an intraperitoneal injection of a 5% *w*/*v* Inactin solution (180 mg/kg). Body temperature was maintained at 37.5 ± 0.5 °C. An arterial catheter connected to a transducer-operated PowerLab system (AD Instruments, Hastings, UK) recorded systemic arterial blood pressure to validate the general condition of the animals. Rats with a mean blood pressure below 70 mmHg were excluded from the study. The abdomen was opened along the midline, and a jejunal (6–15 cm) segment was cannulated, covered with polyethylene wrap, and placed outside the abdomen [56]. The bile duct was cannulated to avoid pancreaticobiliary secretion into the duodenum. 

### 4.4. Perfusion Study

After completion of surgery, ^51^Cr-EDTA was administered IV as a bolus of 75 μCi (0.4 mL), followed by a continuous IV infusion at a rate of 50 μCi per hour (1 mL/h) throughout the experiments. During the first 30 min following surgery, each jejunal segment was single-pass perfused with phosphate-buffered saline (6 mM, pH 6.5, 37 °C) to stabilize cardiovascular, respiratory, and intestinal functions and ^51^Cr-EDTA levels in the circulation. The length and wet tissue weight of each intestinal segment was determined after the experiment. The single-pass perfusion rate was at all times 0.2 mL/min (peristaltic pump, Gilson Minipuls 3, Le Bel, France).

Following the 30 min stabilization period, nine different SPIP experiments were performed under different conditions (Figure 5 and Table 2) with continued monitoring of intestinal mucosal integrity. In two of the nine SPIP experiments, the control solution was perfused for 120 min, with and without luminal addition of melatonin (100 μM) 30 min into the perfusion. Melatonin was added to investigate its effect on the basal integrity of the intestinal mucosa, as melatonin has been reported to reduce the basal duodenal permeability in male Sprague Dawley rats [12].

Seven of the nine SPIP experiments were divided into three periods. Initially, the segment was perfused with the control solution for 45 min to establish baseline conditions in each rat. This was directly followed by a 15 min perfusion of the SDS solution to evaluate the effect on the mucosal leakage and integrity. Finally, the control solution was perfused for 60 min, after which the recovery of the mucosa was evaluated.

Initially, the three-part perfusion experiment was performed with and without pretreatment of the rats with parecoxib (a selective COX-2 inhibitor) to evaluate the basal SDS effect on jejunal permeability. All other groups were pretreated with parecoxib as it restores physiologic enteric nerve activity after abdominal surgery [34,35]. Thirty minutes into the 45 min control period (15 min before the start of the luminal SDS perfusion), either melatonin (100 μM) or misoprostol (10 μM) or melatonin (100 μM) plus misoprostol (10 μM) were added to the perfusate solutions. The drugs were thereafter present in the lumen during the remaining 90 min of the three-part perfusion. 

In two other experiments, teduglutide (100 μg bolus, followed by infusion 33 μg/h) was administered IV at *t* = 0 min and melatonin (20 mg/kg) at *t* = 30 min. (No further drug combination experiments with teduglutide were investigated as no effect was observed by administering it individually.) The luminal and/or IV administrations of melatonin, misoprostol, and teduglutide were given to evaluate their effect on SDS-induced mucosal injury.

All experimental periods started with a rapid filling (<30 s) of the whole jejunal segment with the perfusate (about 1.5 mL for a 10 cm segment). The intestinal segment and perfusates were kept at 37 °C, and all outgoing perfusate was collected and weighed at 15 min intervals. Blood samples (<0.3 mL) were drawn from the femoral artery at the start (*t* = 0 min) and at the end (*t* = 120 min) of the perfusions. The blood samples were centrifuged (5000× *g*, 3 min at 4 °C) within 10 min, and the plasma was analyzed for ^51^Cr activity.

For histological examinations, the perfused jejunal segments were excised immediately after the 120 min perfusion, rinsed with tap water, and fixated in formaldehyde. In two separate experiments, histology was also performed on the intestinal segments directly following the 15 min perfusion of the SDS solution (i.e., no recovery period) with and without the luminal combination of melatonin (100 μM) and misoprostol (10 μM).

### 4.5. Determination of Blood-to-Lumen Jejunal ^51^Cr-EDTA Clearance

All luminal perfusates and blood plasma were analyzed at 0 and 120 min for ^51^Cr activity (cpm) in a gamma counter (1282 Compugamma CS, Pharmacia AB, Sweden). A linear regression analysis of the plasma samples was made to obtain a corresponding plasma value for each perfusate sample. The blood-to-lumen CL_Cr-EDTA_ was calculated using Equation (1) [57]: (1)$${\mathrm{CL}}_{\mathrm{Cr}-\mathrm{EDTA}}=\frac{C_{perfusate\;}\;\times \;Q_{in}}{C_{plasma\;}\;\times \;tissue\;weight\;}\times 100$$
where *C_perfusate_* and *C_plasma_* are the activities in the perfusate and plasma (cpm/mL), respectively, and *Q_in_* is the flow rate (mL/min) in the segment. CL_Cr-EDTA_ is expressed as mL/min/100 g wet tissue weight. CL_Cr-EDTA_ values over time from 0 to 120 min are presented from the nine perfusion experiments. In the evaluation of CL_Cr-EDTA_ over time, CL_Cr-EDTA_ was normalized against the average value in the 45 min control period. The area under the CL_Cr-EDTA_ time curve between 45 and 120 min (CL_AUC_) and the maximum CL_Cr-EDTA_ concentration (CL_max_) were then calculated using noncompartmental analysis in GraphPad Prism version 7.0 for windows (La Jolla, CA, USA.)

### 4.6. Histology

Three cross segments of the jejunum—two from regions close to the resection margins and one from the specimen midpoint—were excised for further investigation. The sections were routinely prepared and cut into 3 μm slices and stained with hematoxylin and eosin and Alcian blue–PAS (pH 2.5). An experienced gastrointestinal pathologist assessed the specimen in a blinded fashion using a light microscope. The following criteria were investigated: epithelial alterations and disruption or detachment of the epithelium; damage of the mucus layer or altered mucus production; bleeding or edema; and signs of acute inflammation (as determined by the presence of neutrophils). All tissues were graded as: (−) no effect, (+) small effect, and (++) clear effect.

### 4.7. Statistical Analysis

Based on previous studies, a sample size of six rats was used in the CL_Cr-EDTA_ experiments and four in the histology experiments [13,22]. All descriptive CL_Cr-EDTA_ statistics are presented as the mean ± standard deviation (SD) or standard error of the mean (SEM). All values from the nine groups were compared using a one-way ANOVA with a post-hoc Tukey’s multiple comparison test. Differences were considered to be statistically significant at *p*-value < 0.05. 

## Figures and Tables

**Figure 1 ijms-21-06771-f001:**
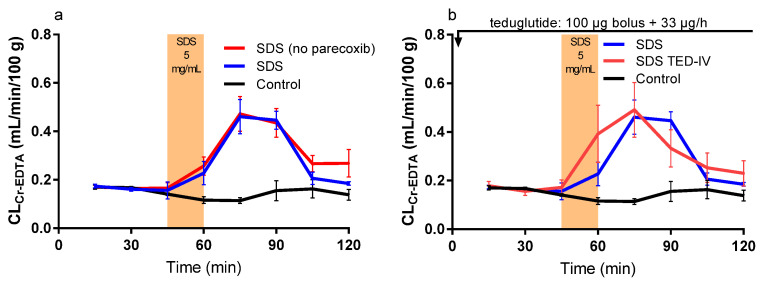
The blood-to-lumen ^51^Cr-EDTA clearance (CL_Cr-EDTA_) between 0 and 120 min for the control buffer solution with or without sodium dodecyl sulfate (SDS) at 5 mg/mL added to the luminal perfusate between 45 and 60 min. (**a**) The effect of intravenously pretreating rats with 10 mg/kg parecoxib 30 min before the start of the SDS experiment. (**b**) The effect of intravenously treating rats with a 100 μg teduglutide bolus at time 0, followed by a 33 μg/h teduglutide infusion during the whole SDS experiment.

**Figure 2 ijms-21-06771-f002:**
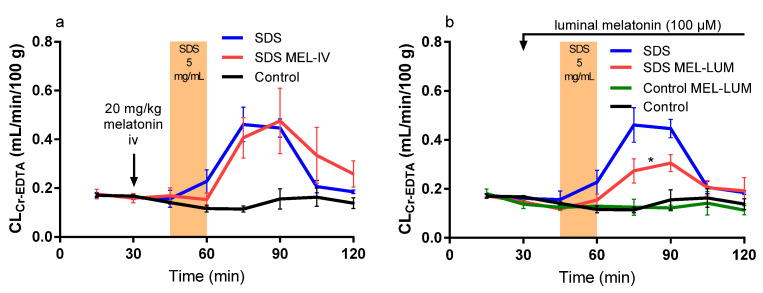
The blood-to-lumen CL_Cr-EDTA_ between 0 and 120 min for the control buffer solution with or without SDS at 5 mg/mL added to the luminal perfusate between 45 and 60 min. (**a**) The effect of intravenously treating rats with a 20 mg/kg melatonin bolus 30 min before the start of SDS exposure. (**b**) The effect of luminally treating rats with melatonin (100 μM) between 30 and 120 min in the control and SDS experiments. The * represents a significant decrease in CL_Cr-EDTA_ exposure of the luminal melatonin compared to SDS alone.

**Figure 3 ijms-21-06771-f003:**
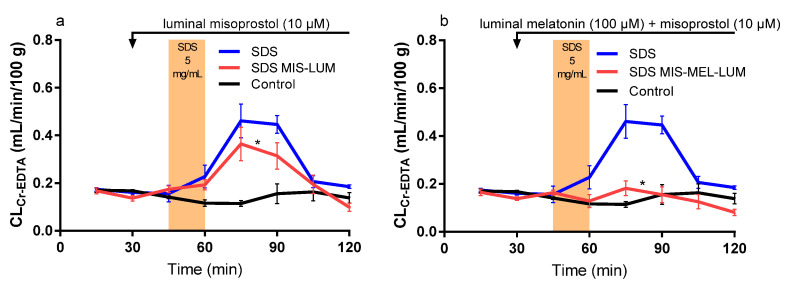
The blood-to-lumen CL_Cr-EDTA_ between 0 and 120 min for the control buffer solution with or without SDS at 5 mg/mL added to the luminal perfusate between 45 and 60 min. (**a**) The effect of luminally treating rats with misoprostol (10 μM) between 30 and 120 min in the SDS experiment. (**b**) The effect of luminally treating rats with melatonin (100 μM) and misoprostol (10 μM) between 30 and 120 min in the SDS experiment. The * represents a significant decrease in CL_Cr-EDTA_ exposure of the luminal drug treatments compared to SDS alone.

**Figure 4 ijms-21-06771-f004:**
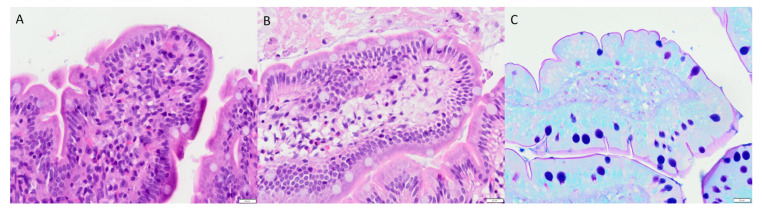
Two pictures (**A**,**B**) of hematoxylin and eosin-stained villous tips, and one (**C**) picture showing Alcian blue–PAS (pH 2.5)-stained mucus (original magnification ×400). (**A**) The villous tip contain a moderate, densely packed infiltrate of macrophages, lymphocytes, plasma cells, and eosinophils. No neutrophils were present. (**B**) An edema is shown with an increase of the intercellular stroma and a loose presence of inflammatory cells and fibrocytes. Lymphatic and blood vessels are dilated. The epithelium is intact. (**C**) A continuous small layer of magenta-positive mucus at the luminal apical face of the enterocytes. The goblet cells display a blue-purple mucus. Apoptotic cells and macrophages containing small mucus droplets are seen under the epithelium. Experimental groups: (**A**) melatonin 100 mM, (**B**) misoprostol 10 mM, and (**C**) control.

**Figure 5 ijms-21-06771-f005:**
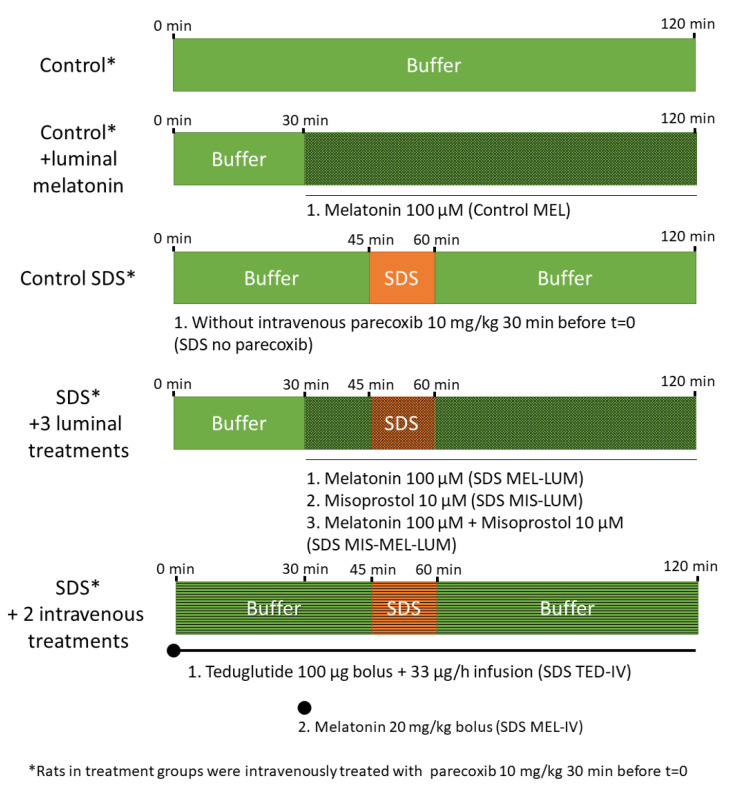
The luminal compositions, conditions, and treatments of the nine different experiments. The jejunum of rats (n = 6) was single-pass perfused for 120 min with a pH 6.5 saline buffer solution (Control) with or without adding the mucosal irritant SDS at 5 mg/mL between 45 and 60 min. At 30 min before the start of the SDS experiments, all groups except one (Control SDS without parecoxib) were intravenously (IV) pretreated with 10 mg/kg parecoxib, a selective cyclooxygenase-2 inhibitor. The three experimental drugs—melatonin, misoprostol, and teduglutide—were luminally and/or intravenously administered, individually or in combination, to the rats in order to evaluate their effects on the SDS-induced mucosal damage. A mesh illustrates that drug is added to the luminal perfusate (buffer and/or SDS), and stripes illustrates that drug is added intravenously while no change is made to the luminal perfusate.

**Table 1 ijms-21-06771-t001:** The area under the blood-to-lumen CL_Cr-EDTA_ time curve between 45 and 120 min (CL_AUC_), and maximum CL_Cr-EDTA_ (CL_max_) in the jejunal perfusate of the nine different single-pass intestinal perfusion experiments (Table 2 and Figure 5). Three experimental drugs—melatonin (MEL), misoprostol (MIS), and teduglutide (TED)—were luminally (LUM) and/or intravenously (IV) administered, individually or in combination, to rats in order to evaluate their effects on SDS-induced mucosal damage. Differences in CL_AUC_ and CL_max_ compared to the control solution or the SDS solution are presented. The mark * represents a significantly lower CL_AUC_/CL_max_ of the luminal combination of melatonin and misoprostol compared to melatonin and misoprostol alone. The CL_AUC_ and CL_max_ values were compared using a one-way ANOVA with a post-hoc Tukey’s multiple comparison test.

Experiments	CL_AUC_ (mL/100 g)	CL_max_ (mL/min/100 g)	CL_AUC_ and CL_max_ Significant Different from:	
			Control	SDS
Control	10.3 ± 1.6	0.16 ± 0.02	-	Yes
Control MEL-LUM	9.6 ± 1.1	0.14 ± 0.02	No	Yes
SDS no parecoxib	24.7 ± 3.0	0.47 ± 0.06	Yes	No
SDS	22.7 ± 1.8	0.46 ± 0.04	Yes	-
SDS TED-IV	25.1 ± 4.6	0.49 ± 0.09	Yes	No
SDS MEL-IV	23.7 ± 4.8	0.48 ± 0.10	Yes	No
SDS MEL-LUM	16.4 ± 2.1	0.31 ± 0.04	Yes	Yes
SDS MIS-LUM	18.0 ± 2.6	0.36 ± 0.05	Yes	Yes
SDS MEL-MIS-LUM	10.7 ± 1.6*	0.18 ± 0.03*	No	Yes

**Table 2 ijms-21-06771-t002:** Conditions and treatments of the nine different experiments. The jejunum of rats (n = 6) was single-pass perfused for 120 min with a pH 6.5 saline buffer solution (Control) with or without adding the mucosal irritant SDS at 5 mg/mL between 45 and 60 min. Before the start of the jejunal perfusions, all groups except one were intravenously (IV) pretreated with 10 mg/kg parecoxib, a selective cyclooxygenase-2 inhibitor. The three experimental drugs—melatonin (MEL), misoprostol (MIS), and teduglutide (TED)—were luminally (LUM) and/or intravenously (IV) administered, individually or in combination, to the rats in order to evaluate their effects on SDS-induced mucosal damage.

Experimental Groups	SDS Luminally Added between 45 and 60 min	Drug Treatments (Dose/Concentration)
Control	No	-
Control MEL-LUM		Melatonin lumen (100 μM)
SDS no parecoxib	Yes	-
SDS		-
SDS TED-IV		Teduglutide IV (100 μg bolus + 33 μg/h infusion)
SDS MEL-IV		Melatonin IV (20 mg/kg bolus)
SDS MEL-LUM		Melatonin lumen (100 μM)
SDS MIS-LUM		Misoprostol lumen (10 μM)
SDS MEL-MIS-LUM		Melatonin lumen (100 μM) + Misoprostol lumen (10 μM)

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
