# Peer review of "Prevention of Rat Intestinal Injury with a Drug Combination of Melatonin and Misoprostol"

_ijms, 2020, doi:10.3390/ijms21186771_

Round 1
Reviewer 1 Report
Dahlgren and his colleagues studied the effects of melatonin, misoprostol and teduglutide on acute jejunal injury in rats. They perfuse rat’s jejunum with SDS and evaluate 51Cr-EDTA clearance for 60 minutes after the mucosal insult.
The main result of the study is that pre-treatment with a combination of melatonin and misoprostol can reduce mucosal damage and completely inhibit the increase in 51Cr-EDTA clearance caused by SDS.
The methods are well described, and the discussion supports well the results.
Minor spell and grammar checks need to be made (for example at lines 136-138 there is still some template).
The experimental design is very interesting although the use of only 6 rats per experiment is rather poor as demonstrated by the great standard deviation seen in the teduglutide and melatonin experiments. Maybe it would be better to widen the experimental groups. Of note, it would be interesting to know if the drugs have effects on intestinal permeability in control conditions.
In figure 2b the authors say that luminal melatonin significantly decreases SDS-induces 51Cr-EDTA clearance, but it is not shown in the graph. This also happens in figure 3b. It would be better to show the significance in these graphs too.
In the histology part of the results, the authors show clear signs of edema in the villi after misoprostol exposure, however there are no statements made on the combination of misoprostol and melatonin effects. Does the combination show the same effects? Is there edema after combination treatment? Moreover, a picture of the villi of rats exposed to SDS should be added as a control reference.
In the discussion part, the authors state that teduglutide does not show protective effects. Is it possible that it can have any protective effects with a higher concentration?
Lastly, SDS acts by dysregulating tight junctions. Did the authors investigate the protective roles of these drugs on tight junction proteins? For instance, it would be interesting to see an immunohistochemistry staining or a western blot of these proteins in order to further confirm the protective role of the drugs in study. Also, some speculations about the cellular mechanism that protects the mucosa from injury could be made after these additional experiments.
Reviewer 2 Report
The manuscript shows partial preventing effects of melatonin and misoprostol on the luminal SDS perfusion-induced intestinal injury, and the complete effects of their combination by utilizing rat jejunum in vivo. The results and conclusion are very interesting and valuable to publish, but the manuscript should be revised according to follows.
In the text of Discussion (line 191-192) and Table 2, it was described that all groups except one {SDS (no parecoxib)?}, were intravenously pretreated with parecoxib, but not described what time it was. I found the different descriptions about it. In Figure 1 legend, it was described “30 minutes before the start of SDS experiment”, but in Figure 5 legend, “before the start of the jejunal perfusions.” What were correct?
Does the group “SDS” in all Figures and Tables indicate “SDS (no parecoxib)”? Thus, in Figure 1, “SDS +parecoxib” was correctly “SDS (no parecoxib)”, right?
In table 2 (and possibly in all results), “SDS (no Parecoxib)” was compared with the experimental groups as the control group. However, “SDS +parecoxib” should be the control group, if all experimental groups were pretreated with parecoxib.
In Figure 4, it is unclear what groups these images are acquired from.
Minor:
Line 136 – 138: Didn’t you forget to erase them?
Reviewer 3 Report
This study focuses on an interesting and highly relevant topic. The main novelty of this paper is the pre-treatment with misoprostol and melatonin to reduce the effects of SDS.
Comments:
This study is about the cytoprotective effects of the drugs tested and I feel this needs to be made clearer in the abstract and introduction. The treatments are given before the SDS to prevent/reduce injury. It is not to repair existing injury.
I feel it is confusing at times what the different treatments are and a better job could be done to explain them. It was easier to understand in Figure 1,2 and 3 what the treatments are in terms of what was added when. Figure 5 should be modified to make it more understandable. Better naming of the treatment groups is needed and then used consistently throughout the text.
Line 15: “there are no approved drugs that target the intestinal epithelial barrier”. I think target is the wrong word to use here. Maybe “can repair” is better or “specifically designed to reduce intestinal permeability”.
Line 16: Change why to while
First paragraph of the introduction has only 2 references. More are needed as ref 1 is a review. Ref 2 is about mice not humans. A better reference might be: Barker N. Adult intestinal stem cells: critical drivers of epithelial homeostasis and regeneration. Nat Rev Mol Cell Biol 2014; 15:19–33; PMID:24326621; http://dx.doi.org/10.1038/nrm3721
Section 2.5. Histology
More information on the effects on the treatment groups is needed with more explanation. The two criteria to assess the effectiveness of the treatments are the 51Cr-EDTA clearance and histology but little histology is provided. The methods say the histology was assessed in a blinded fashion for different criteria. Was it scored ? How was the comparison made ?
Are the authors saying that there were no differences between groups except “The only feature that differed between the study groups was a clear edema in the tip of several villi in the two experimental groups exposed to misoprostol for 60 min”. Which groups were these ? Is the iv and luminal addition ?
Line 137-138: Remove the sentence: It should provide a concise and precise description of the experimental results, their interpretation as well as the experimental conclusions that can be drawn.
Figure 4.
- The difference between a and b is not explained in the figure legend. Also it would be nice to see some labels or arrows to identify certain cells in the diagram to show the features mentioned.
- Why is not more of the histology shown such as SDS alone and the damage compared to the other treatment groups. I know there were no damage observed but it would be good to see the image.
Section 4.3:
- more information needed on how animals were housed and fed and mention if fasted.
- Figure 5 it needs to made clearer that all groups but one have parecoxib added. Is that the group Control SDS or does control SDS refer to two groups (+/- parecoxib).
Section 4.5. Determination of blood-to-lumen jejunal 51Cr-EDTA clearance
Was any software in particular used to perform the non-compartmental analysis ?
Table 1:
Why were comparisons made to the SDS (no parecoxib) solution rather than the SDS (+parecoxib) ?
Round 2
Reviewer 2 Report
I thought you revised your manuscript adequately. I recommend the editors to accept the revised version of the manuscript.